# Effect of Heat Treatments on the Corrosion Resistance of a High Strength Mg-Gd-Y-Zn-Zr Alloy

**DOI:** 10.3390/ma15082813

**Published:** 2022-04-12

**Authors:** Hang Xu, Yuan Li, Luoyi Wu, Fulin Jiang, Dingfa Fu, Jie Teng, Hui Zhang

**Affiliations:** Department of Materials Science and Engineering, Hunan University, Changsha 410082, China; xuhang@hnu.edu.cn (H.X.); 13638421696@163.com (Y.L.); tengjie@hnu.edu.cn (J.T.); zhanghui63hunu@163.com (H.Z.)

**Keywords:** magnesium alloys, heat treatment, corrosion resistance, microstructure

## Abstract

Magnesium-rare earth (Mg-Re) alloys are very promising structural materials in lightweight industries, while the poor corrosion resistance limits their widespread application. In this work, to provide insights into the functions of precipitate characteristics on the corrosion behaviors of Mg-Re alloys, the influence of heat treatments on the corrosion behavior of Mg-11.46Gd-4.08Y-2.09Zn-0.56Zr alloy was investigated via an immersion test, electrochemical experiment and scanning electron microscope (SEM). The results showed that the corrosion rate of Mg-11.46Gd-4.08Y-2.09Zn-0.56Zr alloy specimens decreased by 17.58% and 20.44% after T5 and T6 heat treatment, respectively. In the heat-treated specimens, the corrosion did not extend further into the matrix but formed fine corrosion grooves along the extrusion direction. The improved homogeneity reduced the residual stress and the *β*’ precipitate introduced as a corrosion barrier after T5 and T6 heat treatment reduced the corrosion rate of the studied Mg alloy. In addition, the volume fraction of the long-period stacking-ordered (LPSO) phase decreased after heat treatment, which effectively reduced galvanic corrosion and enhanced the protective effect on the Mg matrix.

## 1. Introduction

Recent years have witnessed a growing research interest in magnesium (Mg) and its alloys in automotive, aerospace and biomedical fields due to their high specific strength, high specific stiffness, damping performance and good biocompatibility [1,2,3,4,5,6]. However, further applications are limited by their heat resistance and creep resistance [7,8]. The addition of rare earth elements in Mg alloys can improve the microstructure, mechanical properties, corrosion resistance, creep resistance and oxidation resistance of alloys at room temperature and high temperature [9,10,11]. Because of the important role of LPSO microstructure, Mg-Re-Zn-Zr alloy has good high-temperature strength and creep resistance [12,13]. However, rapid corrosion rate and local corrosion pattern hinder the wide applications of such rare-earth Mg alloys.

In Mg-RE-Zn-Zr alloys, it is generally believed that the galvanic corrosion of alloys is caused by the heterogeneous LPSO structure [14,15,16,17]. For instance, Li et al. [18] noted that the LPSO phase in the as-cast Mg-Zn-Y alloy acted as a microcathode, thereby promoting the corrosion process. The galvanic effect between the LPSO phase (cathode) and the α-Mg matrix (anode) determined the corrosion rate of the as-cast alloy. Wang et al.’s [19] further research indicated that the increase in the interface between 18R and 14H LPSO structures will decrease the corrosion resistance of solution-treated alloys because the complex crystallographic characteristics of the interface between 18R and 14H LPSO structures led to the high dissolution rate of the grain boundary in the transition region. Recently, Wu et al. [20] found that the hydrogenation of the Mg-Gd-Zn-Zr alloy would replace the massive surface LPSO phase with a new rare earth hydride phase Gd_0.5_Y_0.5_H_1.98_, thus improving the corrosion resistance. Ma et al. [21] also found that alloying Al and Sn elements effectively improved the protection of the oxide film. After heat treatment, the content of MgO on the corroded surface increased, the matrix was separated from the corroded medium and the alloy corrosion rate decreased. However, there are few studies on the effect of precipitate characteristics under particular aging conditions on corrosion behaviors of rare-earth Mg alloys.

In this work, to provide insight into the function of precipitate characteristics on the corrosion behaviors of rare earth Mg alloys, different microstructures were first tailored by T5 and T6 treatments. Then, the corrosion resistance of the Mg-Gd-Y-Zn-Zr alloy specimens with various microstructures was investigated via an immersion test, electrochemical experiment and scanning electron microscope (SEM).

## 2. Materials and Methods

In the present study, the Mg-11.46Gd-4.08Y-2.09Zn-0.56Zr (wt.%) alloy was adopted. The alloy was obtained by melting high-purity (99.9%) Mg, Zn, Y, Mg-30.6Gd and Mg-30.33Zr (wt.%) master alloys in a mild steel crucible under the protection of a mixed atmosphere of SF_6_ (10 vol%) and CO_2_ (bal.). Then, as shown in Figure 1, the cast ingot was firstly extruded into Mg alloy bars. The extrusion temperature was approximately 400 °C and the extrusion ratio was approximately 10. After being ground and polished, the specimens were etched via a solution with a composition of 5 mL of acetic acid, 2.5 g of picric acid, 100 mL of ethyl alcohol and 10 mL of distilled water. The optical microstructure of extruded Mg-11.46Gd-4.08Y-2.09Zn-0.56Zr was obtained by AXIOVERT40 metallography microscope and is provided in Figure 1. The actual composition (Table 1) of the studied alloy was measured by JY-38S and VARIAN715 inductively coupled plasma emission spectroscopy (ICP) in the Analysis and testing Institute of Hunan Rare Earth Metal Research Institute.

As shown in Figure 1, the Mg-11.46Gd-4.08Y-2.09Zn-0.56Zr alloy specimens were either directly aged at 200 °C or aged after solution treatment at 500 °C for 10 h. The samples with peak hardness (designated as T5 and T6 specimens, respectively) were adopted for corrosion testing. The metallographic samples were corroded after cutting, grinding and polishing. The etch agent was 2.5 g picric acid +5 mL glacial acetic acid +10 mL distilled water +100 mL ethanol. After corrosion, the microstructure morphology was observed under a 4XCE optical electron microscope. The grain size was measured by ImageJ software. A JSM-6060LA scanning electron microscope (SEM) equipped with a ZEISS EVOMA10 tungsten filament and an energy dispersive spectrometer (EDS) was used to observe the microstructure and corrosion morphology of the samples and analyze the second phase composition. Hardness was measured by the HVS-1000 microhardness tester with a load of 200 kgf and a loading time of 15 s. The average value of 5 points of each sample was taken as the result.

Immersion corrosion tests were performed using a 3.5% NaCl solution saturated with Mg(OH)_2_, and the corrosion rate was evaluated by the weight loss method. Before immersion corrosion, 36.2 g NaCl was dissolved in 1000 mL of distilled water to obtain the initial solution, and then Mg(OH)_2_ of more than 0.5 g was added. The prepared solution was stirred for half an hour to ensure it was uniform and then filtered twice with slow-speed filter paper. The Mg alloy samples used for the immersion test were cut into blocks with 10 × 8 × 3 mm^3^ by wire-electrode cutting. The samples for the immersion test were polished with 400#, 1000# and 2000# sandpaper and successively cleaned with distilled water and acetone. The weight of the tested samples before and after immersion was measured and recorded after drying. The corrosion rate was calculated by the weight loss method according to ASTMG31–12A, and the formula is as follows:(1)$$C_{R}=\frac{K\times W}{A\times t\times \rho }$$
where *C_R_* is the corrosion rate (mm/year), *K* is the constant (8.76 × 10^4^), *W* is the weight loss value (g), *A* is the surface area of the sample (cm^2^), *t* is the immersion time (h), and *ρ* is the density of the sample (*ρ* = 2 g/cm^3^ in this study).

The electrochemical polarization behavior of the studied Mg alloy was measured by the CHI660B electrochemical workstation. The 3.5% NaCl solution saturated with Mg(OH)_2_ was utilized. In this three-electrode system, a saturated calomel electrode and platinum electrode were used as the reference electrode and counter electrode, respectively. Moreover, the sample was a working electrode, which was encapsulated in a polyvinyl chloride (PVC) pipe with resin so that only one surface was exposed as the working surface. After grinding the working face with 1000# and 2000# sandpaper, the tested Mg alloy was cleaned successively with distilled water and acetone. Then, after a 1 h open-circuit potential test, the polarization curve of action potential was tested. The scanning rate was 0.5 mV/s, and the scanning range was open circuit potential (OCP)−500 mV~OCP + 1500 mV. Accordingly, the corrosion rate is also estimated by Equation (2):(2)$$C_{R}=\frac{Ar\times Icorr}{n\times F\times \rho }$$
where *C_R_* is the corrosion rate (mm/year), *Ar* is the atomic mass, *I_corr_* is the corrosion current (A/cm^2^), *n* is the number of electrons transferred by an electrochemical reaction, *F* is the faraday constant (1 F = 26.8 A·h), and *ρ* is the density of the sample (*ρ* = 2 g/cm^3^ in this study).

## 3. Results and Discussions

Figure 2 shows the hardness evolutions at an aging temperature of 200 °C for the extruded and solution-treated Mg-11.46Gd-4.08Y-2.09Zn-0.56Zr alloy. As can be seen in Figure 2, the microhardness of the extruded alloy is approximately 99.9 HV. When the extruded alloy is aged at 200 °C, an obvious age-hardening effect is displayed. The hardness increases slowly in the early stage of aging. The hardness value increases rapidly with the extension of the aging time from 4 h to 36 h and reaches the peak hardness of 119.0 HV. Then, with the further extension of the aging time, the hardness decreases slowly and then tends to be stable for a long time, indicating that the alloy has good heat resistance.

The extruded alloy was solution-treated at 500 °C for 10 h and then aged at 200 °C (Figure 1). The aging hardening curve is similar to that of the as-extruded alloy, although the initial hardness of the solution-treated specimen is lower with a value of approximately 82.1 HV. The reason is that the grains grow after the solution treatment, and the dislocation density decreases, resulting in a decrease in hardness. After aging for 42 h, the peak hardness is maintained, and good thermal stability is achieved for a long duration. Compared with the T5 state, it takes a longer time to reach the peak aging state (T6) and the peak hardness value is slightly lower. On the other hand, the aging hardening effect of T6 is more obvious, and the microhardness of T6 increases by 44.7% compared to the solution-treated alloy. This is because the supersaturated solid solution formed in the matrix after solution treatment is decomposed during aging and more of the second phase is precipitated.

The optical microstructures of the Mg-11.46Gd-4.08Y-2.09Zn-0.56Zr alloy under T5 and T6 states are presented in Figure 3. The grain size is also given in the upper right corner. Fully dynamic recrystallization of the alloy takes place during the extrusion process, forming a fine recrystallization structure. The second phase is then broken, and streamlined distribution is observed along the extrusion direction. For the T5 specimen, as shown in Figure 3a, the average grain size is 4.13 μm and the grain size is not uniform. The grain size is smaller in the position close to the second phase, while the grain size is larger in the position without the second phase. Although the solid solution of rare-earth atoms in the magnesium matrix blocks the migration of sub-boundaries and inhibits dynamic recrystallization, the LPSO phase can refine grains through the particle stimulated nucleation (PSN) mechanism, so the grains near the LPSO phase are refined [22]. In addition, the broken LPSO phase particles prevent the further growth of recrystallized grains. After aging, the microstructure was more uniform, and the grains grew slightly. As shown in Figure 3b, the grains grow significantly after solution treatment and the average grain size of the T6 alloy is 15.4 μm. A massive second phase is distributed along the grain boundary in the extrusion direction, because recrystallized grains are formed after solution treatment and the LPSO phase is transformed into a nubby LPSO phase. At the same time, the rare-earth-rich particles near the nubby LPSO phase visibly increase.

Figure 4 indicates the SEM diagram and EDS analysis of the Mg-11.46 Gd-4.08Y-2.09Zn-0.56Zr alloy under T5 and T6 states. It can be seen from Figure 4a,b that many fine second phases are uniformly distributed along the extrusion direction in the as-extruded sample. The EDS results at point B are shown in Figure 4d, which is close to the reported composition of the LPSO phase [23]. In addition, cubic phases can be observed at higher multiples. As shown in Figure 4c, its chemical composition is 73.34Mg-8.08Gd-17.33Y-0.75Zn (at.%), which may be a rare-earth-rich phase. After aging at 200 °C for 36 h, its microstructure is similar to that of the extruded alloy, but certain nano-precipitated phases cannot be observed [23,24]. After the T6 treatment, the second phase is still evenly distributed along the extrusion direction, but the LPSO phase was transformed into strips. EDS results of point D (Figure 4h) indicate the chemical composition of 86.91Mg-4.28Gd-3.56Y-5.13Zn (at.%). EDS analysis of the rare-earth-rich phase is shown in Figure 4g, and its chemical composition is 73.34Mg-8.08Gd-17.33Y-0.7Zn (at.%).

Figure 5 shows the corrosion rates (mm/year) of the studied alloys with different heat treatment states in a 3.5% NaCl solution saturated with Mg(OH)_2_ measured by the weight loss method according to Equation (1). The corrosion rates of the as-extruded and T5- and T6-treated samples are 15.07 mm/y, 12.42 mm/y and 11.99 mm/y, respectively. The corrosion rate of the alloy studied in this paper is moderate compared with the results previously studied [25]. The results show that the corrosion rate of the T5 specimen is slightly higher than that of the T6 specimen, while the corrosion rate of the extruded specimens is the highest.

Compared with the as-extruded alloy, the corrosion resistance of the T5- and T6-treated specimens is visibly improved. On the one hand, the microstructure of the extruded magnesium alloy is not uniform, and there is certain residual stress. After heat treatment, the homogeneity of the alloy is improved, and the effect of the non-uniform microstructure on the corrosion resistance is alleviated. On the other hand, the fine and uniform *β*’ phase precipitated after heat treatment can serve as a barrier to corrosion propagation [26,27]. According to previous publications [28,29,30], fine grain size can enhance the corrosion performance of magnesium alloys because a small grain size will produce more grain boundaries, which act as a corrosion barrier. Although the grain size of the T6 alloy is obviously coarser than that of the T5 alloy, the grain boundary acting as a corrosion barrier is reduced, but the aging process after the solution precipitated more *β*’ phase from the supersaturated solid solution. Then, more *β*’ phases acted as corrosion barriers than that of the T5 alloy. Hence, the corrosion rate of the T6 state and the T5 state is similar. In addition, the volume fraction of the LPSO phase decreases after the solution heat treatment, which effectively reduces the galvanic corrosion and enhances the protective effect on the matrix. Figure 5 also shows the macroscopic corrosion morphology after immersion in a 3.5% NaCl solution for 24 h. A chromic acid solution was used to wash away the corrosion products on the sample surface, showing the different corrosion morphologies with different states. It can be found that the corrosion degree of the as-extruded sample is obviously more serious, and many serious local corrosion pits are distributed on the surface. After the aging treatment, the corrosion degree is lighter. Uniform and shallow corrosion pits are observed in T5- and T6-treated specimens. The corrosion degree observed in macroscopic corrosion morphology is consistent with the estimated corrosion rate of immersion weight loss.

To further observe the influence of heat treatment on the corrosion behavior of the studied alloy, the microscopic corrosion morphology under SEM of the immersed samples was observed in Figure 6. Compared to the T5 sample with a smoother surface, the as-extruded alloy displays groove-like pits at a lower magnification, showing deeper corrosion pits. From the high magnification morphology, the corrosion area of the as-extruded specimen is more uniform, while the T5 sample shows an obvious grooved corrosion pit. With the extension of the aging time, the volume fraction of the second phase decreases and the size increases. It can be speculated that the groove-like corrosion pits of the T5 sample distributed along the extrusion direction are due to the preferential corrosion of the electrocoupled corrosion formed between the LPSO phase as the cathode and the magnesium matrix as the anode. However, because the size of the LPSO phase is smaller and the distribution is uniform, the corrosion pits are deeper and more uniform at a higher magnification. As in the T5 sample, the T6 sample is corroded because the cathode’s LPSO phase is distributed along the extrusion direction, and the corrosion pits are also grooved along the extrusion direction (Figure 6e). In the corrosion pits, a nubby LPSO phase and a small, rare-earth-rich phase remain after the magnesium matrix is corroded (Figure 6f), which is consistent with previous reports [20,31]. In addition, by comparing Figure 6d,f, it can be found that the corrosion groove of the T6 sample is narrower, deeper and more evenly distributed than that of the T5 sample, which may be caused by the growth of the cathode’s LPSO phase after solution treatment, and the continuous accumulation of the corrosion process in the Mg matrix near the cathode’s LPSO phase. On the other hand, the main reasons for reducing the corrosion rate of the Mg-11.46Gd-4.08Y-2.09Zn-0.56Zr alloy are improved homogeneity, reduced residual stress and the β’ precipitate introduced as a corrosion barrier after heat treatment. So, the corrosion rates of the T5- and T6-treated materials are similar.

Figure 7a shows the OCP curves of the as-extruded, T5 and T6 samples in a 3.5 wt.% NaCl solution. At the beginning stage, the OCP value of the T5 specimen is approximately −1.659 V_SCE_, which increases rapidly to −1.604 V_SCE_ and decreases rapidly to between −1.648 V_SCE_ and −1.643 V_SCE_, and then the sample is stable after 3600 s immersion. The open-circuit potential curves of the T6 and T5 specimens have similar characteristics, rising rapidly in the early stage and then falling. After that, the curves become stable after dipping for 500 s in a small range. A Mg(OH)_2_ film or passivation is formed in T5 and T6 states at the initial stage of soaking [32]. Due to local surface activation, the formation of new film and the dissolution of the old film reached equilibrium when the Mg(OH)_2_ film was destroyed. Therefore, the open-circuit potential curve showed an increase and then a decrease and tended to be stable. The OCP value of the as-extruded sample fluctuated in a small range and stabilized at −1.627 V_SCE_ after 3600 s. The stable OCP value of the as-extruded samples is approximately 18 mV_SCE_, which is higher than that of the T5 samples. Liu et al. [33] reported that the high stable OCP value of the Mg alloy treated with a solution is due to the formation of more protective surface layers.

Figure 7b shows the potential dynamic polarization curves of the studied Mg alloy samples after a 3600 s open-circuit potential test. It can be found that the anode and cathode branches of the measured polarization curves are asymmetrical. The current density in the anode branch increases much faster than that in the cathode branch, and this phenomenon is also found in other magnesium alloys [34,35,36]. The cathode branch is controlled via the hydrogen evolution reaction, and there is no fuzzy passivation area in the anode branch, indicating the absence of a protective film on the alloy surface. However, due to the negative difference effect and pitting, anodic branches are generally not suitable for fitting analysis. In order to compare the corrosion resistance of the three samples, the corrosion parameters were determined by using the cathode branch of polarization curves. The corrosion current density *I_corr_* is determined by the intersection of *E_corr_* and the cathodic Tafel slope. The fitted corrosion parameters are shown in Table 2. It is generally believed that samples with a lower corrosion current density have better corrosion resistance. The corrosion potential of as-extruded, T5 and T6 specimens are −1.469 V vs. SCE, −1.475 V vs. SCE and −1.521 V vs. SCE, respectively. Moreover, the corrosion currents are 1.126 × 10^−4^ A/cm^2^, 1.815 × 10^−5^ A/cm^2^ and 1.518 × 10^−5^ A/cm^2^, respectively. The results show that the self-corrosion current perceptibly decreases after heat treatment, which reflects the corrosion rate of the studied alloy. According to the self-corrosion current, the performance of corrosion resistance follows the sequences of T6 state > T5 state > as-extruded specimens, which is consistent with the result of weightlessness soaking (Figure 5). In addition, the corrosion rate calculated according to Equation (2) shows the same trend. The results of the as-extruded, T5- and T6-treated samples are 2.20 mm/y, 0.41 mm/y and 0.30 mm/y, respectively. The calculated *C_R_* is approximately one order of magnitude lower compared to the weight loss method. This may be because (1) the presence of Mg^+^ in the corrosion process of magnesium alloy means that Mg = Mg^2+^ + 2e^−^ is only a part of the electrochemical reaction, and the corrosion rate measured by the electrochemical method underestimates the true corrosion rate. (2) The corrosion of magnesium alloys first occurs in the local area, and then slowly expands to cover the entire surface, and the corrosion rate increases accordingly. Therefore, the corrosion rate measured by the short-term corrosion test is likely to be less than that measured by the stable corrosion test, while the electrochemical method is a short-term and early test method compared with the weight loss method. (3) Tafel extrapolation is essentially a transient testing technique that provides a way to measure corrosion rates at a specific time. For magnesium alloys, the immersion test allows for the measurement at any specific time throughout the duration of immersion. In contrast, the corrosion rate measured by immersion is an average over the duration of immersion. According to the cathode branch of polarization curves, the electrochemical reaction under the same conditions is driven by the hydrogen evolution reaction. However, the rate of hydrogen evolution after heat treatment is lower than that of the as-extruded specimen, indicating that T5 and T6 treatments inhibit the cathode process.

## 4. Conclusions

The effect of heat treatments on the corrosion behavior of the Mg-11.46Gd-4.08Y-2.09Zn-0.56Zr alloy was studied. The main conclusions are as follows:

(1) A fine recrystallization microstructure was obtained in as-extruded Mg-11.46Gd-4.08Y-2.09Zn-0.56Zr alloy specimens due to the dynamic recrystallization during the extrusion process. The second phase was also finely broken and presented a streamlined distribution along the extrusion direction. After the T5 treatment, the grain size grew slightly and the volume fraction of LPSO decreased. After the T6 treatment, the grain size was obviously coarser. Meanwhile, the strip LPSO phase was formed, and the rare-earth-rich particles increased significantly.

(2) Based on the evaluations from the weight loss method and the electrode polarization approach, the corrosion rate of Mg-11.46Gd-4.08Y-2.09Zn-0.56Zr alloy specimens decreased obviously after both T5 and T6 heat treatments, and the corrosion rate of T5 sample was slightly higher than that of T6 sample. After the T5 treatment, the corrosion degree was significantly reduced, and the corrosion did not extend further into the matrix but formed fine corrosion grooves along the extrusion direction. The T6 sample showed similar corrosion grooves, and there were massive LPSO phase and rare-earth-rich particles in the corrosion grooves.

(3) The improved homogeneity reduced the residual stress, and the *β*’ precipitate introduced as a corrosion barrier after heat treatment reduced the corrosion rate of the studied Mg-11.46Gd-4.08Y-2.09Zn-0.56Zr alloy. In addition, the volume fraction of the LPSO phase decreased after heat treatment, which effectively reduced the galvanic corrosion and enhanced the protective effect on the Mg matrix.

## Figures and Tables

**Figure 1 materials-15-02813-f001:**
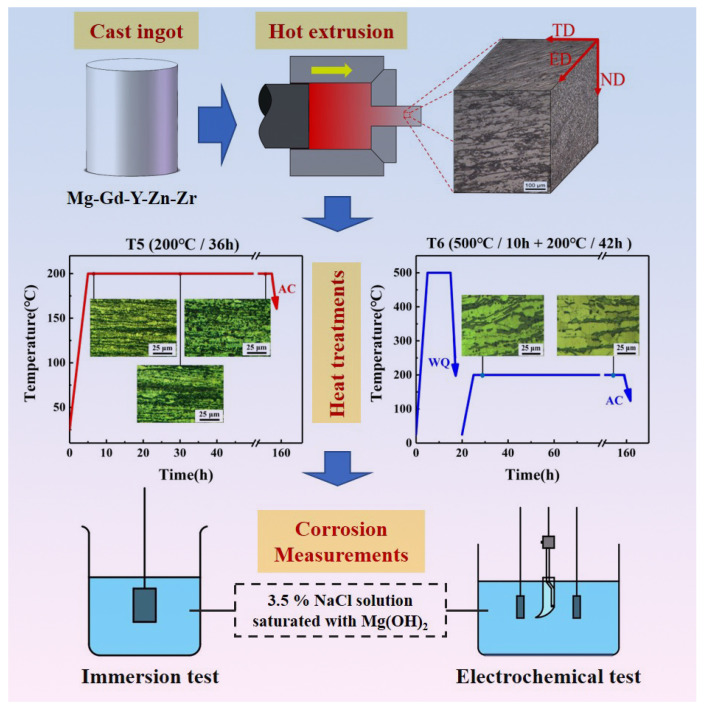
The overall experimental process and the initial microstructure of the studied alloy.

**Figure 2 materials-15-02813-f002:**
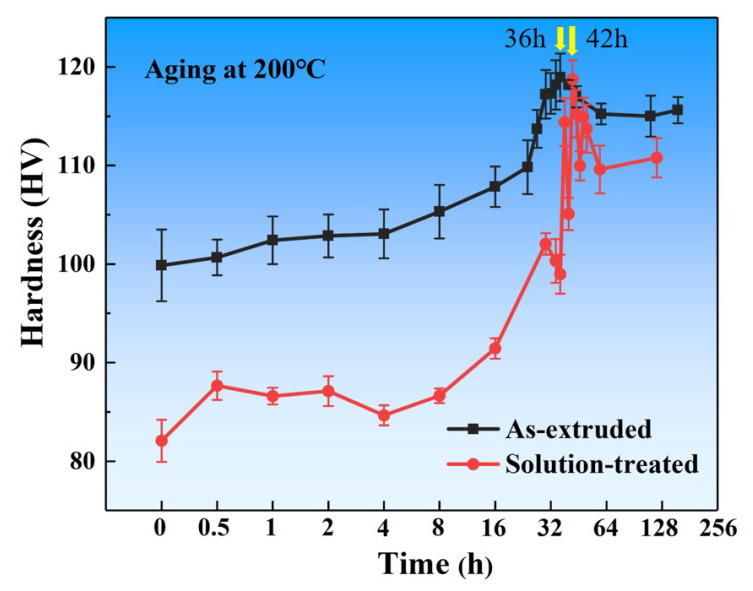
The hardness evolutions at aging temperature of 200 °C for the as-extruded and solution-treated Mg-11.46Gd-4.08Y-2.09Zn-0.56Zr alloy.

**Figure 3 materials-15-02813-f003:**
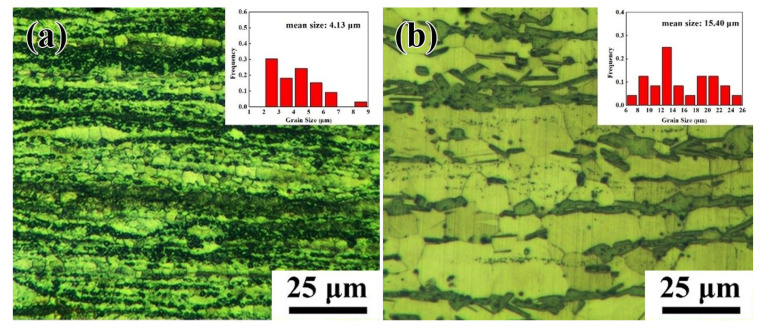
Optical microstructures of the Mg-11.46 Gd-4.08Y-2.09Zn-0.56Zr alloy under (**a**) T5 and (**b**) T6 states.

**Figure 4 materials-15-02813-f004:**
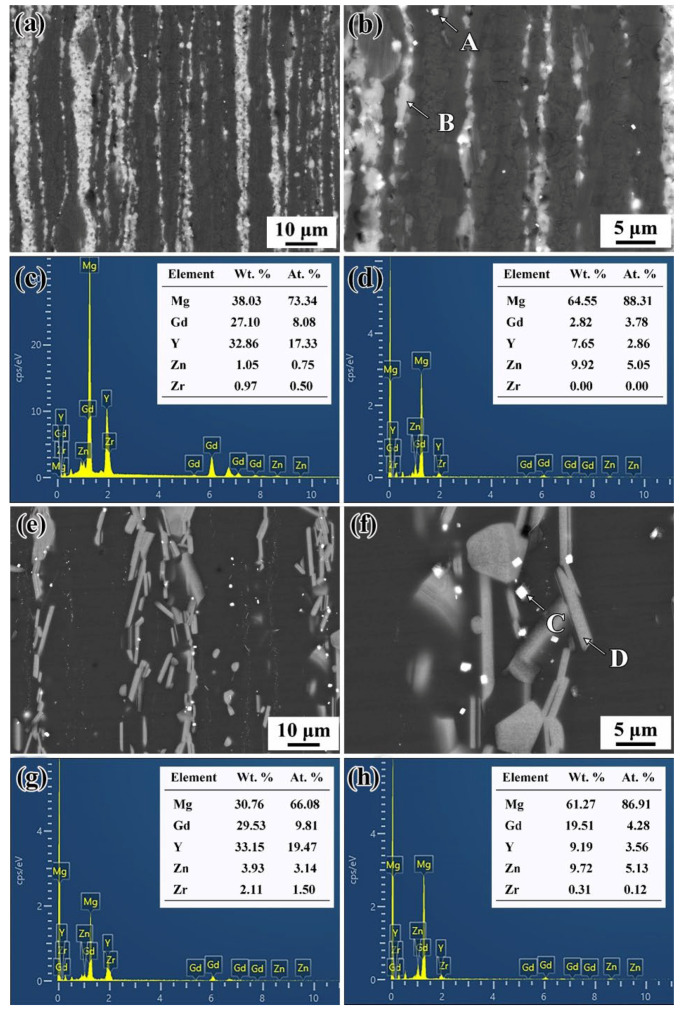
SEM and EDS analysis of the Mg-11.46 Gd-4.08Y-2.09Zn-0.56Zr alloy: (**a**,**b**) SEM results of T5-treated alloy; (**c**,**d**) EDS results corresponding to A and B in (**b**); (**e**,**f**) SEM results of T6-treated alloy; (**g**,**h**) EDS results corresponding to C and D in (**f**).

**Figure 5 materials-15-02813-f005:**
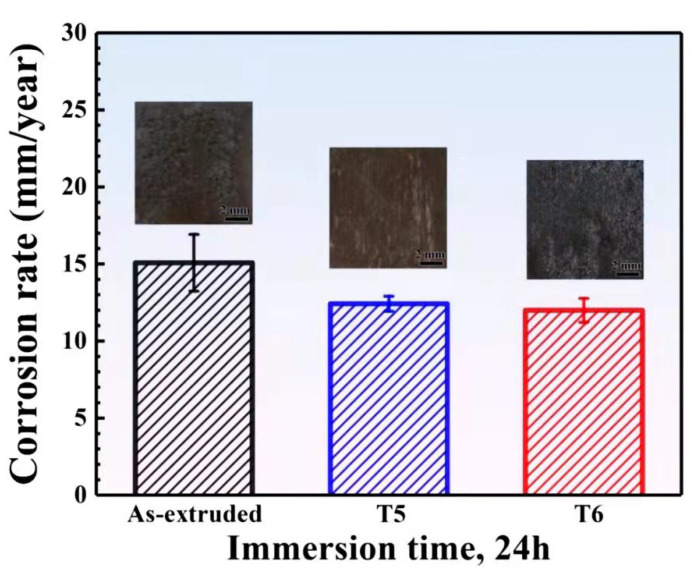
Corrosion rates of the as-extruded, T5- and T6-treated Mg-11.46Gd-4.08Y-2.09Zn-0.56Zr alloys estimated by weight loss method.

**Figure 6 materials-15-02813-f006:**
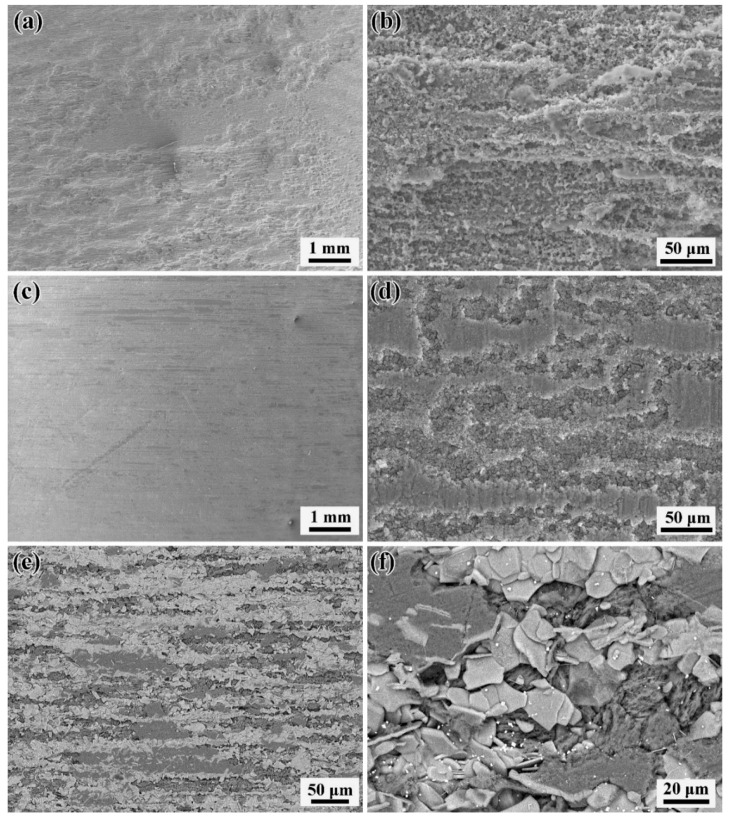
Microscopic corrosion morphology under SEM after immersion in 3.5% NaCl solution saturated with Mg(OH)_2_ for 24 h: (**a**,**b**) As-extruded sample; (**c**,**d**) T5 sample; (**e**,**f**) T6 sample.

**Figure 7 materials-15-02813-f007:**
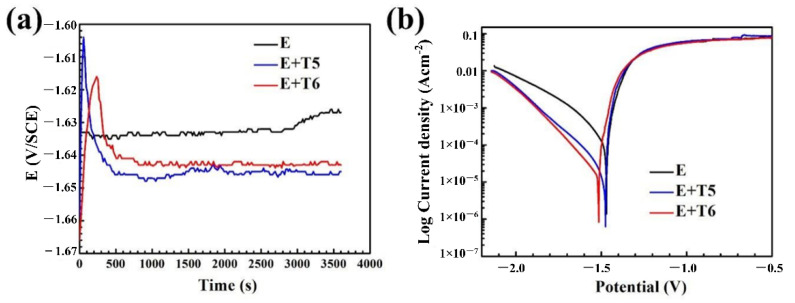
The electrochemical polarization behaviors of as-extruded, T5, T6 specimens in 3.5% NaCl solution: (**a**) Open-circuit potential curve; (**b**) potentiodynamic polarization curves.

**Table 1 materials-15-02813-t001:** Chemical compositions of the studied Mg-Gd-Y-Zn-Zr alloy.

	Gd	Y	Zn	Zr	Mg
wt.%	11.46	4.08	2.09	0.56	81.81
at.%	2.07	1.30	0.91	0.17	95.55

**Table 2 materials-15-02813-t002:** The results of OCP, corrosion potential and corrosion current of as-extruded, T5 and T6 specimens.

Sample	OCP (V vs. SCE)	*E_corr_* (V vs. SCE)	*I_corr_* (A/cm^2^)
E	−1.627	−1.469	1.106 × 10^−4^
E + T5	−1.645	−1.492	2.089 × 10^−5^
E + T6	−1.643	−1.521	1.518 × 10^−5^

## Data Availability

Not applicable.

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
