# Peer review of "Effect of Heat Treatments on the Corrosion Resistance of a High Strength Mg-Gd-Y-Zn-Zr Alloy"

_materials, 2022, doi:10.3390/ma15082813_

Round 1

Reviewer 1 Report

The manuscript entitled "Effect of heat treatments on the corrosion resistance of a high strength Mg-Gd-Y-Zn-Zr alloy”. The authors study the corrosion behavior of an Mg-Gd-Y-Zn-Zr alloy that has been subjected to an extrusion process and subsequent thermal treatment, using morphological and electrochemical techniques.

The manuscript presents considerable problems that need to be clarified before uploading it again for evaluation, such as:

  • Eq 1 used to determine the corrosion rate in this manuscript is not suitable for the system under study because Magnesium alloys present localized corrosion, as you describe in the results and many more works in the literature. The equation is only valid for uniform corrosion.
  • The scanning rate used in the polarization curves (0.5 mV/s) is higher than that reported in most corrosion works and recommended by experts in the area, which is 0.16mV/s. Although higher speeds can be used, these must be justified, especially when it is decided to use a speed higher than the recommended one, it is because the system is very stable and the definition in the signal is not lost.
  • Include a figure showing polarization curves at different scanning rates to demonstrate that the scanning used in this manuscript is valid.
  • Clarify how a measure of hardness is related to a thermal property. This is due to the claims made regarding Figure 2.
  • Eliminate Figure 5 as it is constructed based on an erroneous approximation.
  • The description of the polarization curves (Fig. 7b) is not precise, review the description and improve it. The figure clearly shows an evident and stable passive branch starting at -1.2V, indicating that the material does form a passive layer.
  • Describe in the manuscript how the corrosion current densities in Table 2 were obtained.
  • Regarding the Icorr, no differences can be seen in the graphs that support this difference of one order of magnitude reported in Table 2. Passive systems, such as the one observed in Figure 7b, the icorr is iqual to ipass since it is the controlling process.
  • In the conclusions, the corrosion resistance part should be written, considering the scope of the techniques used to evaluate this property.

Also, I recommend using another electrochemical technique to describe your system's corrosion resistance adequately.

Author Response

Dear Editor and Reviewers,

Thank you very much for your kind works and the reviewers’ significant comments concerning our manuscript with the title " Effect of heat treatments on the corrosion resistance of a high strength Mg-Gd-Y-Zn-Zr alloy" (materials-1657501). The careful comments are very valuable and helpful for improving our paper. We have studied the comments carefully and tried our best to make corrections which we hoped meet with approval. The new version of the manuscript is uploaded, and revised portions are also marked in red in the marked-up manuscript. The responses to the reviewers’ comments are shown as following:

Reviewers’ comments & our responses:

The manuscript entitled "Effect of heat treatments on the corrosion resistance of a high strength Mg-Gd-Y-Zn-Zr alloy”. The authors study the corrosion behavior of an Mg-Gd-Y-Zn-Zr alloy that has been subjected to an extrusion process and subsequent thermal treatment, using morphological and electrochemical techniques.

The manuscript presents considerable problems that need to be clarified before uploading it again for evaluation, such as:

Eq 1 used to determine the corrosion rate in this manuscript is not suitable for the system under study because Magnesium alloys present localized corrosion, as you describe in the results and many more works in the literature. The equation is only valid for uniform corrosion.

Response: Thank you very much for your careful comments. On the one hand, we would like to add a hydrogen evolution method, but our campus is currently under controlling and quarantine due to the COVID-19, which makes it impossible for us to conduct the additional experiment. On the other hand, the corrosion rate determined by Eq. (1) indicated the overall functions and was still helpful for evaluating localized corrosion. We searched articles in related fields and found that the weight loss method is still used by some researchers, such as https://doi.org/10.1016/j.corsci.2021.109746. In addition, when the corrosion products on the metal surface are easy to remove, and the metal body will not be damaged by removing the corrosion products, the weight loss method is often used. We strictly followed the above two requirements in the experiment, which greatly improved the accuracy of the experimental results.

The scanning rate used in the polarization curves (0.5 mV/s) is higher than that reported in most corrosion works and recommended by experts in the area, which is 0.16mV/s. Although higher speeds can be used, these must be justified, especially when it is decided to use a speed higher than the recommended one, it is because the system is very stable and the definition in the signal is not lost. Include a figure showing polarization curves at different scanning rates to demonstrate that the scanning used in this manuscript is valid.

Response: Thank you so much for your careful comments. Your comments are nice for guiding our further works. In this work, the scanning rate used in the polarization curves (0.5 mV/s) is a reference to previous work (https://doi.org/10.1016/j.corsci.2021.109746). And we have verified that the present scanning rate could obtain stable results. Due to the realistic conditions, we cannot do additional experiments. We are very sorry about this.

Clarify how a measure of hardness is related to a thermal property. This is due to the claims made regarding Figure 2.

Response: Thanks for your comment. It is known that the formation of precipitates during aging treatment would increase the hardness of such Mg alloys (precipitation strengthening). And the peak hardness would be obtained at critical aging time. After the peak hardness, the strength of most Mg alloys would decrease rapidly due to the fast coarsening of precipitates. But the hardness of the alloy in this work (Fig. 2) decrease slightly to a stable value after the peak value. Hence, the thermal stability (of hardness when holding at high temperature for a long time) is better than most Mg alloys, for instance AZ series alloys [1, 9].

Eliminate Figure 5 as it is constructed based on an erroneous approximation.

Response: Many thanks for the careful comment. I'm sorry that we can't carry out any additional experiments now due to the COVID-19, so we can't use another method to describe the corrosion performance. We're sorry about that. For the integrity of the article, we must keep Figure 5. Such results have been used in other publications for evaluating corrosion rate. In addition, we made some adjustments to Figure 5 to improve the clarity of macroscopic corrosion morphology.

The description of the polarization curves (Fig. 7b) is not precise, review the description and improve it. The figure clearly shows an evident and stable passive branch starting at -1.2V, indicating that the material does form a passive layer.

Response: Many thanks for your careful comments. We reviewed the manuscript and tried our best to improve it. The revisions are marked in red in the manuscript. The anodic branch of the polarization curves is steeper than the cathodic branch, i.e., the current density increases faster in the former. The anodic branch represents the dissolution of the Mg matrix, and the cathodic branch represents the hydrogen evolution reaction. It can also be seen that there are no obvious passivation zones in the polarization curves, which indicated that no protective films formed on these sample surfaces (https://doi.org/10.1016/j.corsci.2021.109746). This is consistent with literature reports, and we made some adjustments in the manuscript to avoid misunderstanding.

Describe in the manuscript how the corrosion current densities in Table 2 were obtained.

Regarding the Icorr, no differences can be seen in the graphs that support this difference of one order of magnitude reported in Table 2. Passive systems, such as the one observed in Figure 7b, the icorr is iqual to ipass since it is the controlling process.

Response: Thank you very much for your careful comments. The corrosion current density Icorr is determined by the intersection of Ecorr and the cathodic Tafel slope. It is generally believed that samples with a lower corrosion current density have a better corrosion resistance. On the other hand, the changing trend of Icorr in different samples reflects the improvement of corrosion properties after heat treatment, which supports our conclusion to a certain extent.

In the conclusions, the corrosion resistance part should be written, considering the scope of the techniques used to evaluate this property.

Response: Thank you so much for your good suggestion. In the revised paper, the conclusions have been improved according to your suggestion, where the techniques was included.

Also, I recommend using another electrochemical technique to describe your system's corrosion resistance adequately.

Response: Many thanks for your suggestion. It is a great pity that COVID-19 has prevented us from conducting any additional experiments. We are very sorry for this and hope that the corrections will meet with approval.

We appreciate for the editor’s and reviewers’ warm works earnestly, and hope that the correction will meet with approval. Please fell no hesitate to tell us if there are further comments.

Thank you so much again.

Kind regards,

Fulin Jiang

Reviewer 2 Report

Review of paper no. materials-1657501 titled Effect of heat treatments on the corrosion resistance of a high strength Mg-Gd-Y-Zn-Zr alloy by H. Xu et al.

This is an interesting article that studies the corrosion behavior of the Mg-RE alloy. The alloy was heat-treated by two different heat treatments (T5 and T6). The results show that the corrosion resistance of the heat-treated alloy has been improved significantly. The paper is publishable subject to revision.

1.The authors say that a long-period stacking order phase (LPSO) has been observed in the alloy. Nevertheless, diffraction data are not presented. Please, include the XRD patterns of the materials, both before and after the heat treatment to support your claim.

2.Two corrosion methods are presented in the paper (weight loss and electrode polarization). However, the corrosion rate (CR) is calculated from weight loss data only. Please, estimate the CR from the corrosion current too. Compare the results with the weight loss method.

3.The corrosion rates of the T5 and T6 treated materials were nearly identical (Fig. 5). However, the materials had markedly different microstructures (Fig. 3). Why is the effect of different microstructures not significant? Provide some discussion.

4.Please compare your results with similar alloys studied previously. See https://doi.org/10.1016/j.jma.2020.08.002 for the collection of Mg alloy corrosion rates.

5.The observance of Mg(OH)2 needs to be experimentally verified. Please, include either an XRD or an EDS analysis of corrosion products in the paper.

6.A hydrogen evolution method could have been included to study the corrosion rate of the Mg alloy. However, this is only a suggestion for future work.

Author Response

Dear Editor and Reviewers,

Thank you very much for your kind works and the reviewers’ significant comments concerning our manuscript with the title " Effect of heat treatments on the corrosion resistance of a high strength Mg-Gd-Y-Zn-Zr alloy" (materials-1657501). The careful comments are very valuable and helpful for improving our paper. We have studied the comments carefully and tried our best to make corrections which we hoped meet with approval. The new version of the manuscript is uploaded, and revised portions are also marked in red in the marked-up manuscript. The responses to the reviewers’ comments are shown as following:

Reviewers’ comments & our responses:

Review of paper no. materials-1657501 titled Effect of heat treatments on the corrosion resistance of a high strength Mg-Gd-Y-Zn-Zr alloy by H. Xu et al.

This is an interesting article that studies the corrosion behavior of the Mg-RE alloy. The alloy was heat-treated by two different heat treatments (T5 and T6). The results show that the corrosion resistance of the heat-treated alloy has been improved significantly. The paper is publishable subject to revision.

1.The authors say that a long-period stacking order phase (LPSO) has been observed in the alloy. Nevertheless, diffraction data are not presented. Please, include the XRD patterns of the materials, both before and after the heat treatment to support your claim.

Response: Thank you so much for your careful comments. We would like to add such results, but we are currently under quarantine due to the COVID-19, which makes it impossible for us to conduct additional experiments. In our previous work [20] and other publications [13, 14, 22-24], the LPSO has been confirmed by XRD. Alternatively, in the revised paper, the citations of related articles are added to prove the existence of LPSO phase.

2.Two corrosion methods are presented in the paper (weight loss and electrode polarization). However, the corrosion rate (CR) is calculated from weight loss data only. Please, estimate the CR from the corrosion current too. Compare the results with the weight loss method.

Response: Many thanks for your good suggestion. According to your suggestion, we added the corresponding content in the manuscript. We calculate it according to the following formula:

where CR is the corrosion rate (mm/year), Ar is the atomic mass, Icorr is the corrosion current (A/cm2), n is the number of electrons transferred by an electrochemical reaction, F is the faraday constant (1F = 26.8 A·h), and ρ is the density of the sample (ρ=2 g/cm3 in this study). The results of the as-extruded, T5 and T6 treated samples are 2.20 mm/y, 0.41 mm/y and 0.30 mm/y, respectively. This results  also confirm that T5 and T6 heat treatment can improve the corrosion properties of extruded Mg-11.46Gd-4.08Y-2.09Zn-0.56Zr (wt.%) alloy.

3.The corrosion rates of the T5 and T6 treated materials were nearly identical (Fig. 5). However, the materials had markedly different microstructures (Fig. 3). Why is the effect of different microstructures not significant? Provide some discussion.

Response: Thanks for your careful comments. The main reasons for reducing the corrosion rate of Mg-11.46Gd-4.08Y-2.09Zn-0.56Zr alloy are the homogeneity of microstructure, reduced residual stress and the introduced β' precipitate as a corrosion barrier after heat treatment. The above three effects can be achieved in the process of peak aging. On the other hand, the decrease of the volume fraction of LPSO phase after solution treatment leads to the lower corrosion rate of T6 sample than T5 sample. In the new submission, the interpretations were also added.

4.Please compare your results with similar alloys studied previously. See https://doi.org/10.1016/j.jma.2020.08.002 for the collection of Mg alloy corrosion rates.

Response: Thank you for the information and it is very helpful. The corrosion rate of the alloy studied in this paper is moderate compared with the results previously studied (https://doi.org/10.1016/j.jma.2020.08.002). The above useful paper has been cited in the revised paper. T5, T6 heat treatment process is very common in industrial production, so this study is of great help to guide industrialization.

5.The observance of Mg(OH)2 needs to be experimentally verified. Please, include either an XRD or an EDS analysis of corrosion products in the paper.

Response: Thank you very much for your careful comments. We also would like to add such results, but due to the COVID-19, we cannot conduct additional experiment. In order to make up for this deficiency, related articles are added to prove the existence of Mg(OH)2.

6.A hydrogen evolution method could have been included to study the corrosion rate of the Mg alloy. However, this is only a suggestion for future work.

Response: Many thanks for your suggestion. It is a great pity that COVID-19 has prevented us from conducting any additional experiments. So that we were unable to further improve our article within the due date. We are very sorry for this and hope that the corrections will meet with approval.

We appreciate for the editor’s and reviewers’ warm works earnestly, and hope that the correction will meet with approval. Please fell no hesitate to tell us if there are further comments.

Thank you so much again.

Kind regards,

Fulin Jiang

Reviewer 3 Report

This paper studies the effect of the heat treatment on the corrosion resistance of a high strength Mg-Gd-Y-Zn-Zr alloy. Generally, Mg-based alloys are the research interest, due to their lightweight, while the poor corrosion resistance limits their wide applications. The present paper is interesting and have a novelty in the work, however, to be accepted for publication the following comments need to be addressed

  • Minor English changes are required in the revised manuscript

Abstract

  • In the abstract, the aim of this paper is clear. The most significant results should be presented here in numbers, to what content the corrosion rate was changed?‎

Introduction

  • The introduction section needs to be improved by citing new and related articles of the current journal.

Material and experimental details

  • The authors should add the procedures of sample preparations for microstructure analysis in Figure 1.
  • The specifications (model, company, city, country) of the used tools and equipment should be added.

Results

  • Figure 5 needs to be improved. Highly resolution Micrographs should be added.

Otherwise, the results are well organized and discussed.

Author Response

Dear Editor and Reviewers,

Thank you very much for your kind works and the reviewers’ significant comments concerning our manuscript with the title " Effect of heat treatments on the corrosion resistance of a high strength Mg-Gd-Y-Zn-Zr alloy" (materials-1657501). The careful comments are very valuable and helpful for improving our paper. We have studied the comments carefully and tried our best to make corrections which we hoped meet with approval. The new version of the manuscript is uploaded, and revised portions are also marked in red in the marked-up manuscript. The responses to the reviewers’ comments are shown as following:

Reviewers’ comments & our responses:

This paper studies the effect of the heat treatment on the corrosion resistance of a high strength Mg-Gd-Y-Zn-Zr alloy. Generally, Mg-based alloys are the research interest, due to their lightweight, while the poor corrosion resistance limits their wide applications. The present paper is interesting and have a novelty in the work, however, to be accepted for publication the following comments need to be addressed.

Minor English changes are required in the revised manuscript.

Response: Thanks for your careful comments. We have checked the English carefully with the help of our colleague who is very good at English. The detailed revisions are shown in the new submission.

Abstract: In the abstract, the aim of this paper is clear. The most significant results should be presented here in numbers, to what content the corrosion rate was changed?‎

Response: Thank you so much for the nice comments. We agree with your idea and the corresponding statements have been added in the revised manuscript.

Introduction: The introduction section needs to be improved by citing new and related articles of the current journal.

Response: Many thanks for your suggestion. We have added relevant references in the revised manuscript as you suggested.

Material and experimental details: The authors should add the procedures of sample preparations for microstructure analysis in Figure 1. The specifications (model, company, city, country) of the used tools and equipment should be added.

Response: Thanks for your careful comments. We are sorry for our mistakes. We have added the procedures of sample preparations for microstructure analysis in Figure 1 to the revised manuscript.

Results: Figure 5 needs to be improved. Highly resolution Micrographs should be added. Otherwise, the results are well organized and discussed.

Response: Many thanks for the careful comment. We have replaced the figures in Figure 5 with higher image quality.

We appreciate for the editor’s and reviewers’ warm works earnestly, and hope that the correction will meet with approval. Please fell no hesitate to tell us if there are further comments.

Thank you so much again.

Kind regards,

Fulin Jiang

Round 2

Reviewer 1 Report

The manuscript has been improved. I recommend your acceptance to be published.

Author Response

Thank you very much for your kind works.

Reviewer 2 Report

The authors improved their paper. However, the corrosion rate calculated from the corrosion current is presented in the cover letter only. It should be given in the paper itself. Furthermore, the calculated CR (0.3 – 2.2 mmpy) is about one order of magnitude lower compared to the weight loss method (Fig. 5). At least some discussion is necessary. It is not enough to say that the corrosion rate “follows the same trend” (line 284).

Author Response

Many thanks for your careful suggestion. We have added the explanations in the revised paper. After reviewing the literatures, we found that the difference of an order of magnitude between electrochemical method and weight loss method also appeared in previous works (https://doi.org/10.1002/adem.200310405, https://doi.org/10.1002/adem.200600221, https://doi.org/10.1016/j.corsci.2008.04.010, https://doi.org/10.1016/S0010-938X(96)00172-2). These may be because: (1) The presence of Mg+ in the corrosion process of magnesium alloy means that Mg = Mg2+ + 2e- is only a part of the electrochemical reaction, and the corrosion rate measured by electrochemical method underestimates the true corrosion rate. (2) The corrosion of magnesium alloys firstly occurs in the local area, and then slowly expands to cover the entire surface, and the corrosion rate increases accordingly. Therefore, the corrosion rate measured by the short-term corrosion test is likely to be less than that measured by the stable corrosion test, while the electrochemical method is a short-term and early test method compared with the weight loss method. (3) Tafel extrapolation is essentially a transient testing technique that provides a way to measure corrosion rates at a specific time. For magnesium alloys, the immersion test allows measurement at any specific time throughout the immersion time. In contrast, the corrosion rate measured by immersion is an average over immersion time. Related statements are also marked in the new manuscript.
